# Elemental Analysis, Phytochemical Screening and Evaluation of Antioxidant, Antibacterial and Anticancer Activity of *Pleurotus ostreatus* through In Vitro and In Silico Approaches

**DOI:** 10.3390/metabo12090821

**Published:** 2022-08-31

**Authors:** Vartika Mishra, Sarika Tomar, Priyanka Yadav, Shraddha Vishwakarma, Mohan Prasad Singh

**Affiliations:** 1Centre of Biotechnology, University of Allahabad, Prayagraj 211002, India; 2Institute of Nuclear Medicine & Allied Sciences (INMAS), Defence Research and Development Organisation (DRDO), Delhi 110054, India; 3Centre of Bioinformatics, University of Allahabad, Prayagraj 211002, India

**Keywords:** oyster mushroom, bioactive compounds, anticancer, antioxidant, antibacterial, spectroscopy, molecular docking

## Abstract

Oyster mushrooms form an integral part of many diets owing to their characteristic aroma, delicious taste and nutraceutical value. In this study, we examined oyster mushrooms by direct arc optical emission spectroscopy for the presence of various biologically important elements. Furthermore, we screened phytochemicals present in *Pleurotus ostreatus* by applying GC-MS. Additionally, the antioxidant, antibacterial and anticancer activities of the ethanolic extract of *Pleurotus ostreatus* were studied. Moreover, we docked the phytochemicals and examined their binding affinities with EGFR, PR and NF-κB proteins, which are overexpressed in breast cancer. The elemental analysis showed the presence of Fe, K, Na, Ca, Mg, Cr and Sr in the spectrum. Moreover, GC-MS data revealed the presence of 32 biologically active compounds in oyster mushrooms. The ethanolic extract displayed remarkable free radical scavenging activity (~50%) against DPPH. The mushroom has shown promising antibacterial activity against both Gram-positive (*S. aureus*) and Gram-negative bacteria (*Pseudomonas*
*aeruginosa*, *Proteus vulgaris* and *Proteus mirabilis*). The present study also revealed that oyster mushrooms possess significant anticancer activity. The ethanolic extract inhibited the growth and proliferation of MCF-7 cells. It also induced cell shrinkage, membrane blebbing and nuclear fragmentation, resulting in apoptosis of malignant cells. The molecular docking analysis showed that ligand 15 (Linoleic acid ethyl ester), ligand 27 (Ergosta-5,7,9(11),22-tetraen-3-ol, (3. beta.,22E), ligand 28 (Stigmasta-5,22-dien-3-ol, acetate, (3. beta.,22Z), ligand 30 (Ergosta-5,7,22-Trien-3-Ol, (3. Beta.,22E) and ligand 32 (gamma. Sitosterol) exhibited better binding affinities with EGFR, PR and NF-κB proteins. This result provides a strong ground for confirmation of the in vitro anticancer effect of *Pleurotus ostreatus*. From the present in vitro and in silico studies, it can be concluded that *Pleurotus ostreatus* is a useful source of essential elements and reservoir of bioactive compounds which confer its significant antioxidant, antibacterial and anticancer properties.

## 1. Introduction

Edible mushrooms have been used to enrich taste and fulfil nutritional needs across cultures since the dawn of civilization. They have been found to be a major source of carbohydrates, proteins, fats, fibers and minerals, making them a diet of choice. Their aroma, texture and nutritive value earned them names such as the food of God by the Romans and the elixir of life by the Chinese [1]. These natural luxuries contain plenty of important bioactive compounds, e.g., ergosterol, beta glucan, levostatin and triterpenes, which impart to them potential antioxidant, anti-allergic, antidiabetic, immunomodulatory, antineoplastic and antimicrobial properties [2]. Among the many varieties of edible mushrooms, three are notable for their extensive use—white button mushrooms (*Agaricus bisporus* L.), oyster mushrooms (*Pleurotus ostreatus* L.) and paddy straw mushrooms (*Volvariella volvacea* L.) [3]. Among these three, *Pleurotus ostreatus* has recently garnered the attention of scientists for its significant therapeutic potential.

*Pleurotus ostreatus* (oyster mushroom) is the second most consumed and cultivated species of mushroom in the world. It is rich in trace elements essential for our body and is a reservoir of macromolecules such as α-glucans, β-glucans, lentinan, lipopolysaccharides, resveratrol, Cibacron blue affinity purified protein, concanavalin A and many more. The bioactive molecules present in it confer various therapeutic properties, e.g., hypocholesterolemia; free radical scavenging activity and antioxidant, anti-atherogenic, anti-tumor, immunomodulatory and antibacterial effects [2]. The antioxidant role of *Pleurotus ostreatus* has been well documented (EC_50_ = 4.42 mg/mL) and plays a decisive role in combating oxidative stress, DNA damage, carcinogenesis and cellular damage related to aging [4]. Oyster mushrooms were also found to be effective against both Gram-positive (MIC against *Staphylococcus aureus* = 6 mg/mL) and Gram-negative bacteria (MIC against *Pseudomonas aeruginosa* = 8 mg/mL) [5]. The antibacterial role of these edible natural products becomes important as there has been a growing incidence of antimicrobial resistance due to over-the-counter use of antibiotics. Recently, *Pleurotus ostreatus* has gained the attention of scientists for its effective anti-tumor properties. Extracts of *Pleurotus ostreatus* have been found to inhibit the proliferation of breast cancer cells (IC50 = 4.5 μg/mL), induce apoptosis, impedes angiogenesis and consequently arrest the process of metastasis. Components of *Pleurotus ostreatus*, e.g., beta glucan, have been found to display notable immunomodulatory effects, thus acting as an adjuvant in cancer immunotherapy [6,7].

Considering all the above factors, in the present study, we analyzed *Pleurotus ostreatus* by direct current arc optical emission spectroscopy for the presence of essential elements. Thereafter, we explored the ethanolic extract of oyster mushroom for the presence of bioactive compounds using gas chromatography–mass spectrometry and evaluated its antioxidant, antibacterial and anticancer activities. The anticancer effect has also been substantiated via an in silico approach by performing molecular docking studies.

## 2. Material and Methods

### 2.1. Spawn Preparation, Cultivation and Harvesting of Mushroom

Spawn is the vegetative mycelium which precedes the cultivation of mushrooms. For the preparation of spawn, half-boiled and air-dried wheat grain was taken as the substrate. We then added CaCO_3_ and CaSO_4_ in the ratio of 3:1. This substrate mixture was added into a sterilized glass bottle and plugged with a cotton plug followed by autoclaving for 35 min. After 48 h, mycelium of *Pleurotus ostreatus* was inoculated in the autoclaved bottles. The bottles were incubated at 25 ± 2 degrees Celsius. After a 25–30-day incubation period, a white cotton-like mycelial structure was seen surrounding the glass bottle.

In the successive stage, sterilized paddy straw was used as a substrate for the cultivation of mushroom. Here, 50 g spawn per 1 kg of wet weight of paddy straw was mixed thoroughly, and then, the mix was filled in rubber-tied polythene bags. Small holes were made in the polythene bags to facilitate proper aeration and the removal of excess water. After 12–15 days, the small pin head structure of the mushroom first appeared and the polythene packaging was removed thereafter. A fully matured fruiting body was seen in between 20 and 25 days. The fruiting body was plucked, collected, chopped and then air dried. The dried *Pleurotus ostreatus* fruiting body was then powdered with a mortar and pestle and stored at −20 degrees Celsius.

### 2.2. Optical Emission Spectroscopy

The powdered form of mushroom was used as the sample for the direct current arc optical emission spectroscopy technique. The overall setup of the spectroscopic technique is shown in Figure 1, comprising a direct current arc system that includes two carbon electrodes, ballast resistance and a fiber optic-linked spectrometer, which is a multi-channel spectrometer for collecting the emitted radiation. The spectrometer has three gratings that cover three separate spectral ranges: 300–500, 500–700 and 700–900 nm. The mouth of the bottom of the carbon electrode was kept filled with a very low amount of powdered sample which was excited as the voltage increased across the electrodes. As a result, there were distinct radiations produced, which were captured using the fiber optic cable and directed to the slit of the multi-channel spectrometer. The strong and developed emitted signals were acquired using Spec Soft MSS 3.0 software in the 300–900 nm spectral region at a resolution of 0.3 nm.

### 2.3. Ethanol Extraction of Herbs

For ethanol extraction, 5 g of powdered *Pleurotus ostreatus* was stirred with 100 mL of ethanol in a Soxhlet apparatus for 24 h. The ethanol extract of *Pleurotus ostreatus* (EEPO) was then evaporated by placing it on a hot plate for 8 h. The dried extract of herb was stored at 4 °C for further experiments.

### 2.4. GC-MS Analysis

The EEPO was analyzed using GC-MS for the identification of various phytochemicals. The analysis was carried out using the Perkin-Elmer Clarus 680 system (Perkin-Elmer, Inc., Shelton, CT, USA) equipped with a fused silica column, packed with an Elite-5MS capillary column (30 m in length × 250 μm in diameter × 0.25 μm in thickness). Pure helium gas (99.99%) was used as the carrier gas at a constant flow rate of 1 mL/min. For GC-MS spectral detection, an electron ionization energy method was adopted with a high ionization energy of 70 eV (electron Volts), 0.2 s scan time and fragments ranging from 40 to 600 m/z. An injection quantity of 1 μL was used (split ratio 10:1), and the injector temperature was maintained at 250 °C (constant). The column oven temperature was set at 50 °C for 3 min, then raised at 10 °C per min up to 280 °C, and the final temperature was increased to 300 °C for a period of 10 min. The contents of phytochemicals present in the test samples were identified based on comparison of their retention time (min), peak area, peak height and mass spectral patterns with those spectral databases of authentic compounds stored in the National Institute of Standards and Technology (NIST) library.

### 2.5. Antioxidant Assay

We used 2,2-diphenyl-1-picryl-hydrazyl-hydrate (DPPH) to evaluate the free radical scavenging ability of phytochemicals present in the EEPO according to the protocol prescribed by Choi, et al. [8]. For a brief period, 1 mL of the ethanolic extract of *Pleurotus*
*ostreatus* (EEPO) at varying concentrations (25, 50, 100, 250, 500 and 1000 µg/mL) and a standard (ascorbic acid) were mixed with 2 mL (0.02%) of DPPH solution in ethanol. The mixture was then incubated in the dark for 30 min, and the absorbance was measured at 517 nm in the spectrophotometer (6800 UV-VIS spectrophotometer), using ethanol as a blank. The percentage scavenging of DPPH radicals was calculated using the following formula:% DPPH scavenging activity = (Ac − (As − Ab)/Ac) × 100
where Ac is the absorbance of the control, Ab is the absorbance of the blank and As is the absorbance of the sample.

### 2.6. Tested Bacterial Organisms

In total, one Gram-positive (*Staphylococcus aureus* ATCC-25923) and three Gram-negative (*Pseudomonas aeruginosa* ATCC-27853, *Proteus mirabilis* NCIM-2300 and *Proteus vulgaris* NCIM-5266) pathogenic bacterial strains were used in this study for testing the antimicrobial activity of the EEPO. These microorganisms were already available in our lab.

### 2.7. Antibacterial Activity

The antibacterial activity of the EEPO was determined by the agar well diffusion method with slight modification [9]. For a short interval, the ethanolic extracts were dissolved in 3% dimethyl sulfoxide (DMSO) at concentrations of 50, 100, 150 and 200 mg/mL. An overnight culture of each microbial isolate was inoculated to check turbidity in nutrient broth equal to 0.5 McFarland (105 CFU/mL). To assess the antimicrobial activity of the extract, an aliquot of the test culture was swabbed uniformly over the surface of the solidified nutrient agar medium with the aid of a sterile swab. Small wells (5 mm in diameter) were formed in agar plates by a sterile cork borer. At 50, 100, 150 and 200 mg/mL concentrations, 100 µL of the EEPO was loaded into different wells. Ciprofloxacin (5 µg/mL) (for Gram-negative bacteria) and penicillin (10 µg/mL) (for Gram-positive bacteria) dissolved in distilled water were used as test microorganism controls. The plates were incubated at 37 °C for 24 h. After 24 h, the inhibition zone was measured in millimeters (mm) by the antibiotic zone scale. All the experiments were carried out in triplicate and their means were documented.

### 2.8. Cell Culture

The human breast cancer cell line MCF-7 was purchased from the National Centre of Cell Science, Pune, India. The obtained cells were cultured in Minimum Essential Medium (MEM) supplemented with 10% *v*/*v* FBS and 1X penicillin/streptomycin antibiotic in a 75 mm tissue culture flask at 37 °C in a humified atmosphere with 5% CO_2_. The cells were grown up to 80–90% confluency and then sub-cultured using 0.25% Trypsin-EDTA, followed by plating the cells as required for the experiments.

### 2.9. In Vitro Cellular Toxicity/Cell Proliferation

The cellular toxicity in the presence of the EEPO was analyzed using the PrestoBlue viability assay. The effect of the extract on the proliferation and viability of MCF-7 cells was expressed as % cell viability using the formula given below:% Cell viability = [(Abs^treated^ − Abs^control^)/(Abs^control^)] × 100

MCF-7 cells were seeded at a density of 2 × 10^3^ and treated with a different range of concentrations of the extract—i.e., 25, 50, 100, 250, 500 and 1000 µg/mL—and then incubated for 24 and 48 h at 37 °C in 5% CO_2_. Afterwards, the viability of the cells was evaluated using the standard PrestoBlue viability assay protocol [10]. PrestoBlue was reduced by metabolically active cells to give a pink color that provides a quantitative measure of the viability of cells in the presence of the mushroom extract.

### 2.10. Anticancer Activity Using SRB Assay

The sulforhodamine B colorimetric assay was performed to assess the anticancer activity of EEPO extracts on the cultured MCF-7 cells. The malignant cells were seeded at a density of 1 × 10^4^ cells in a 96-well plate and treated with a range of concentrations of EEPO (25, 50, 100, 250, 500 and 1000 µg/mL) in DMEM with 10% FBS in triplicate and incubated at 37 °C in 5% CO_2_ for 48 h. After incubation, we added cold 10% TCA to each well without removing the medium, followed by incubation at 4 °C for 1 h. Subsequently, each well was washed using distilled water 2–3 times and then air dried at room temperature. Next, 100 µL of 0.05% SRB solution was added to each well and incubated for 30 min at room temperature, and the plates were then rinsed quickly four times with 1% acetic acid to remove the unbound dye. This was followed by the addition of 200 µL of 10 mM Tris base solution (pH 10.4) to each well and maintenance in a shaker for 10 min to solubilize the protein-bound dye. Furthermore, the absorbance was measured at 510 nm in a microplate reader.

### 2.11. Nuclear Morphological Examination of Cancer Cells

The cancer cell line was seeded at a density of 1 × 10^4^ cells in a 6-well plate and treated with different concentrations of EEPO (50, 100 and 250 µg/mL) in DMEM with 10% FBS in triplicate and incubated at 37 °C in 5% CO_2_ for 48 h. Afterwards, cells were washed with PBS, followed by fixing in 4% paraformaldehyde in PBS for 10 min at room temperature. DAPI (4,6-Diamino-2-phenylindole) staining was performed to observe the nuclear morphological changes in the treated groups. After permeabilization, the cells were stained with 10 µg/mL DAPI solution for 10 min at 37 °C, followed by washing the cells with PBS 2–3 times. Then, the cells were examined under a fluorescence microscope (Carl Zeiss) at 10× magnification.

### 2.12. Cell Apoptosis Assay

For evaluation of cellular apoptosis using FITC Annexin V-PI, firstly, the cells were seeded at a density of 1 × 10^6^ cells in a 6-well plate and then treated with different concentrations (50, 100 and 250 µg/mL) of the extracts and incubated at 37 °C in 5% CO_2_ for 48 h. After incubation, the cells were harvested to obtain the apoptotic cells and washed with PBS 2–3 times. Then, the cell pellet was stained using Annexin V-FITC (5 µL) and incubated for 10 min followed by staining with 2 µL propidium iodide for 2 min in dark. A flow cytometry analysis was conducted to measure the number of cells that had undergone apoptosis. Propidium-iodide-positive and Annexin-V-negative cells are considered healthy, while those that are PI-negative and Annexin-V-positive are apoptotic, and cells that are both PI-and annexin-V-positive are considered necrotic.

### 2.13. Molecular Docking

Docking was performed with AutoDock Vina (Scripps Research Institute, San Diego, CA, USA) to improve the speed and accuracy of molecular docking with a new scoring function, efficient optimization and multi-threading. For the purpose of docking, we identified 32 potential molecules that have been characterized by GC-MS analysis. These potential molecules were searched for 3D structures in the PubChem database; out of the 32 bioactive molecules, only 25 3D structures were available in PubChem. Further molecular interaction of these 25 structures with three cancer biomarker proteins, namely epidermal growth factor receptor (EGFR) [5GTY], progesterone receptor (PR) [1A28] and NF-kB protein (4OT9), was studied through molecular docking. All the available 3D structures of the compounds and proteins are mentioned in Table 1 below.

### 2.14. Statistical Analysis

Data obtained from the study were analyzed using the GraphPad Prism statistical package program Version 5.0 for Windows (Graph Pad Software, Inc., san diego, California, USA). The values are given as mean and standard error of the mean (SEM). The obtained results were analyzed using a two-way ANOVA, and data are expressed as mean ± standard deviation (SD), *n* = 3. The significance is represented as * *p* < 0.05, ** *p* < 0.01.

## 3. Result

### 3.1. Optical Emission Spectroscopy

The baseline correction method was used to process the raw spectral data with the help of Origin 9 software. The curve fitting approach in Gaussian mode was used to estimate the peak characteristics, including areal intensity and width and area of the spectral profile. Three spectra of samples were recorded, and the intensities derived from the curve fitting method of the persistent lines were averaged to evaluate the accuracy of the results. The averaged intensities of the persistent and strong lines of the detected elements were compared to the averaged intensities of the persistent and strong lines of the most abundant elements to estimate the intensity ratio of the identified elements. The outcomes have been reported as the average value and standard error.

The recorded spectrum is depicted in Figure 2 with a variety of spectral lines of varying intensities. The position of the spectral line reveals the identity of the elements, while the intensity of the lines reveals the quantity of the detected element present in the mushroom. Identification of these atomic lines from the generated spectrum was performed using the NIST spectral database. The persistent lines obtained in the spectrum of the sample of the mushroom have identified elements such as iron (358.2, 373.4, 374.5, 421.3 and 438.1 nm), potassium (404.3, 766.3 and 769.7 nm), magnesium (416.4 nm), chromium (417.7 nm), calcium (422.4, 440.1, 443.2, 445.2, 526.9, 559.4, 610.2, 612.1, 616.1, 643.5 and 670.4 nm), strontium (430.4 nm) and sodium (589.2 nm). Most of the persistent lines of neutral atoms that emerge from the ground state during direct current arc excitation are neutral atoms. The transitions from low-lying upper excited states to the ground state often occur during the low-energy excitation process as direct current arc excitation. Moreover, the spectral region of 380–400 nm was observed to be of bands of CN molecules as the carbon in the rods and biochemicals of the samples dissociates on thermal heating and releases a significant amount of carbon vapor, which combines with the surrounding nitrogen and produces the CN bands. The ratios of the intensity of the iron to calcium (I_358.2_/I_422.4_), potassium to calcium (I_404.3_/I_422.4_), magnesium to calcium (I_416.4_/I_422.4_), chromium to calcium (I_417.4_/I_422.4_), calcium to calcium (I_422.4_/I_422.4_) and strontium to calcium (I_430.4_/I_422.4_) were evaluated from the obtained intensity and found to be 0.14 ± 0.06, 0.15 ± 0.008, 0.47 ± 0.02, 0.27 ± 0.01, 1 ± 0.00 and 0.11 ± 0.007, respectively.

### 3.2. Chemical Profiling of Pleurotus ostreatus

The gas chromatography–mass spectrometry chromatogram shows the presence of 32 peaks corresponding to the bioactive compounds present in the ethanolic extract of *Pleurotus ostreatus* (EEPO). The observed dominant compounds on the basis of % peak area and bioactivity were linoleic acid ethyl ester (32.78%), hexadecanoic acid ethyl ester (21.42), (E)-9-octadecenoic acid ethyl ester (13.75), octadecanoic acid, ethyl ester (4.68), 3-cyclopentyl propionic acid,2-dimethyl amino ethyl ester (4.49), benzyl di ethyl-(2,6-xylylcarbamoylmethyl)-ammonium (3.82), ergosta-5,7,22-trien-3-ol, (3. beta.,22e)-(3.07), 2-(diethylamino)-n-(2,6-dimethylphenyl) ace (2.83), pentadecanoic acid and ethyl ester (2.32). The detected active principles along with their retention time (RT), concentration (peak area %), area and biological functionalities are presented in Table 2 and Figure 3.

### 3.3. Antioxidant Activity of Pleurotus ostreatus

The ethanolic extract of *Pleurotus ostreatus* (EEPO) was evaluated and screened for its free radical scavenging activity using the DPPH assay. The outcome of the scavenging potential of the EEPO is depicted in Figure 4. Our findings suggest that there was an increase in the free radical scavenging activity of the EEPO with the corresponding increase in the concentration of extract with a maximum at 1000 µg/mL (~50% inhibition). However, it was also ascertained that the free radical scavenging activity of the EEPO was comparatively lower than that of ascorbic acid.

### 3.4. Antimicrobial Activity of Pleurotus ostreatus

The antibacterial activity of the EEPO against the tested bacteria (Gram-positive and Gram-negative) was examined using the agar well diffusion method. It was found that at 200 mg/mL, the EEPO elicited the most promising antibacterial activity against both the Gram-positive and Gram-negative bacterial strains. The result documented in Table 3 shows that the inhibition zones of the EEPO (200 mg/mL) were 29.6 ± 0.2, 27.4 ± 0.2, 24.4 ± 0.3 and 16.4 ± 0.2 against *Staphylococcus aureus*, *Proteus vulgaris*, *Pseudomonas aeruginosa* and *Proteus mirabilis*, respectively. It can also be inferred from Figure 5 that the EEPO showed concentration-dependent inhibition against *S. aureus*, *Proteus vulgaris* and *Pseudomonas aeruginosa*, while in the case of *Proteus mirabilis*, the antibacterial activity was concentration-independent.

### 3.5. Pleurotus ostreatus Inhibits the Proliferation of Breast Cancer Cells

The effect of the EEPO was evaluated on the growth and proliferation of breast cancer cells. The MCF-7 cells were treated with increasing concentrations of EEPO (25, 50, 100, 250, 500, 1000 µg/mL) for 24 and 48 h. As Figure 6 indicates, the *Pleurotus ostreatus* extract markedly decreased the proliferation of MCF-7 cells in a dose- and time-dependent manner. Furthermore, in order to confirm the findings of the PrestoBlue viability assay, an SRB assay was carried out in MCF-7 cells since some phytochemicals are known to give false positive results. We found comparable inhibition of cell proliferation in the SRB assay as well. The results also indicated similar statistical data representing a significant anti-proliferative capacity of mushroom extract over the period of 48 h.

### 3.6. Effect of Pleurotus ostreatus on Cellular Morphology

To check the effect of the EEPO on cellular morphology, we examined the changes in MCF-7 cells with different concentrations (50, 100 and 250 µg/mL) of the extract. After incubation for 48 h, the morphological changes in apoptotic cells were detected using fluorescent DAPI staining. As observed in Figure 7, *Pleurotus ostreatus* extract treatment resulted in several morphological features associated with apoptotic cells such as cytoplasmic membrane blebbing, shrinkage of cells and nuclear fragmentation, whereas normal MCF-7 cells showed intact stretched morphological structures. It was also found that alterations in the cellular structures varied in a concentration-dependent fashion, e.g., there was increase in MCF-7 cells with an apoptotic phenotype with the corresponding increase in the concentration of the extract.

### 3.7. Apoptotic Potential of Pleurotus ostreatus

Cellular apoptosis in MCF-7 cells was observed using the Annexin V/PI double staining flow cytometry assay as depicted in Figure 8. The results of the Annexin V/PI double staining assay demonstrated that apoptosis of MCF-7 cells was observed after treatment with 100 µg/mL of EEPO for 48 h. The population of apoptotic cells was found to be significantly greater in the treatment groups as compared to the control. It was observed that the apoptotic cell population significantly increased to 13.3 ± 0.1% when compared with untreated cells (7.8 ± 0.1%). Other than this, a significant increase in early and late apoptotic cells was observed in the treatment group as compared to untreated cells. The increase in apoptosis is an indication that the EEPO can revert the anti-apoptotic hallmark of cancer to normal and can thus suppress the malignancy to a significant extent.

### 3.8. Molecular Docking

Docking of GC-MS-characterized compounds was performed with three cancer target proteins screened by a literature review by using the AutoDock Vina program. Each compound was docked with each cancer target protein, namely EGFR, NF-kB and progesterone receptor. The results of the docking study are presented in Table 4 in the form of the binding energies corresponding to the best fit between the ligands and the targeted proteins. The present study deciphered that ligand 15, ligand 27, ligand 28, ligand 30, ligand 32, ligand 31, ligand 29, ligand 10 and ligand 22 show a binding energy above 7.0 kcal/mol with respect to EGFR protein. Similarly, ligand 15, ligand 29, ligand 30, ligand 22 and ligand 27 presented a binding energy above 6.0 kcal/mol with respect to NF-kB protein. Progesterone receptors are an important biomarker of hormone-positive breast cancer. When the phytochemical compounds were docked with progesterone receptor proteins, it was found that ligand 15, ligand 28, ligand 27, ligand 1, ligand 6, ligand 10, ligand 29, ligand 30, ligand 31 and ligand 17 had the maximum binding tendency with the receptor, with binding energies greater than 6.0 kcal/mol. In addition to this, Table 5 presents the amino acid residues involved in the interaction of the three best fit ligands with target proteins.

## 4. Discussion

Oyster mushrooms form an important part of many diets owing to their texture, characteristic aroma and nutritional value. They have been found to be low in cholesterol and fat content while rich in proteins, thus making them a suitable food supplement for those aspiring to be fit and healthy [32]. Nature has bestowed this edible food with many medicinal properties, e.g., hypocholesterolemia, free radical scavenging, antioxidant, antiatherogenic, anti-tumor, immunomodulatory and antibacterial effects [2]. In light of this, we focused our study to investigate the elemental composition, bioactive compounds and various biological activities of this product.

We screened *Pleurotus ostreatus* powder for the presence of trace elements using direct current arc optical emission spectroscopy. From the recorded spectrum, we identified the presence of biologically important elements, e.g., Fe, K, Mg, Ca, Na, Cr and Sr. The observed result was similar to that obtained earlier by Melinda Fogarasi et al., who deciphered the elemental composition of *Pleurotus ostreatus* grown in Romania using an atomic absorption spectrometer [33]. These elements can serve us by compensating for deficiencies and consequent malnutrition—e.g., Ca can be important in the maintenance and formation of bone, K and Na play a role in maintaining osmotic balance, Fe participates in the biosynthesis of Hb and Mg is a cofactor for a variety of enzymes [34]. However, the presence of Sr and chromium leads to potential toxicological concern [35]. Further studies are needed to determine their effective concentrations and probable negative role, if any.

GC-MS analysis of the EEPO demonstrated the presence of 32 compounds. The peak area and retention time as well as significant biological roles attributed to these bioactive compounds have been listed in Table 2. The analysis revealed that linoleic acid ethyl ester (32.78%), hexadecanoic acid ethyl ester (21.42%) and (E)-9-octadecenoic acid ethyl ester (13.75%) were detected as major volatile compounds in the extract. Linoleic acid ethyl ester has been linked to hypocholesterolemic, anti-arthritic, antihistaminic, anti-coronary, anti-eczemic, anti-acne, antimicrobial and anticancer activities. Previous studies have attributed potent antioxidant and anticancer activities to hexadecanoic acid ethyl ester and (E)-9-octadecenoic acid ethyl ester [36,37,38]. These molecules with their various functions confer significant biological attributes to mushrooms which were further explored via in vitro activity.

Owing to the presence of the above bioactive compounds, oyster mushrooms showed significant DPPH free radical scavenging activity at selected concentrations. The ethanol extract was found to have ~50% free radical scavenging potential at a concentration of 1 mg/mL. This antioxidant activity exhibited by the EEPO was significantly higher than that of the ethanol extract of *Pleurotus ostreatus* mycelium as reported by Vamanu, et al. [39]. The elimination of free radicals from our body can help in curbing the process of aging and preventing cardiovascular and inflammatory diseases, cataracts, carcinogenesis and many more ailments. Thus, oyster mushrooms can serve as a better substitute to synthetic antioxidants which are reported to have deleterious consequence to human health.

In addition to antioxidant activity, *Pleurotus ostreatus* has shown potent antibacterial properties in our study. The EEPO inhibited both Gram-positive (*S. aureus*) and Gram-negative bacteria (*Pseudomonas*
*aeruginosa*, *Proteus vulgaris* and *Proteus mirabilis*) in an effective manner. When compared with the effect of methanolic extracts on *S. aureus* and *P. aeruginosa* as reported by M. Fogarasi, et al., ethanolic extracts of oyster mushroom have shown better inhibitory potential. This is important as there is presently a growing worldwide concern with respect to antimicrobial resistance because of excessive use and misuse of antibiotics. Mushroom in the diet can aid in treatment and helps people to recover from bacterial infection in less time.

Furthermore, *Pleurotus ostreatus* has been previously reported to possess anticancer effects. Our study further confirmed the anti-tumor role of EEPO. The results of the PrestoBlue viability assay and SRB assay demonstrated that oyster mushroom extract significantly inhibited the growth and proliferation of MCF-7 cells. Cancer cells are characterized by evasion of apoptosis. However, our study showed that on treatment with the ethanolic extract, MCF-7 cells underwent significant apoptosis in a dose-dependent manner. This was further verified by morphological examination of the cells undergoing apoptosis by using fluorescent DAPI staining. These MCF-7 cells were seen to have undergone shrinkage, membrane blebbing and nuclear fragmentation, all of which contribute to apoptosis. 

Molecular docking studies were performed to understand the probable mechanistic details underlying the above in vitro results. The docking results showed that several ligands can effectively bind with MCF-7 biomarkers, namely EGFR, PR and NF-kB, and can potentially alter their conformations. It was found that linoleic acid ethyl ester (32.78%), which was obtained as a major compound in the GC-MS analysis, had the best fit with respect to all three cancer target proteins. The docking result also revealed that linoleic acid ethyl ester, Ergosta-5,7,9(11),22-tetraen-3-ol, (3. beta.,22E), Stigmasta-5,22-dien-3-ol, acetate, (3. beta.,22Z), Stigmast-5-en-3-ol, oleate and Ergosta-5,7,22-Trien-3-ol, (3. Beta.,22E) bind to the receptor tyrosine kinase domain of EGFR and can thus effectively suppress its functioning. The overexpression of EGFR facilitates PR function, which then effectively promotes breast cancer progression. This binding of bioactive ligands with EGFR and PR can suppress their cross-talk, thus inhibiting the downstream signaling cascade and resulting in suppression of the growth and proliferation of MCF-7 cells [40]. Linoleic acid ethyl ester and other potential compounds with a high binding energy against MCF-7 biomarkers can thus be explored further for their anticancer activity. More experimental findings and advanced bioinformatics tools should be employed in further studies to decipher the exact mechanistic details underlying the above found in vitro and in silico results.

## 5. Conclusions

Oyster mushrooms, with their beneficial elemental composition, are a reservoir of bioactive compounds which confer on them their significant antioxidant, antibacterial and anticancer properties. The in silico approach here has revealed that several bioactive compounds, e.g., linoleic acid ethyl ester and others, can bind to the receptor tyrosine kinase domain of EGFR protein, progesterone receptor and NF-kB target proteins. These findings suggest that these molecules can act as anticancer compounds. However, there is still a need to carry out further detailed in vitro and in silico research studies taking these molecules into consideration. We need to assess the mechanistic details with respect to signaling pathways elicited or inhibited by these molecules to suppress cancer cells. Thus, the present work can be of immense help to researchers who are striving to find a solution to the most complex problems that humans are faced with within nature and natural products.

## Figures and Tables

**Figure 1 metabolites-12-00821-f001:**
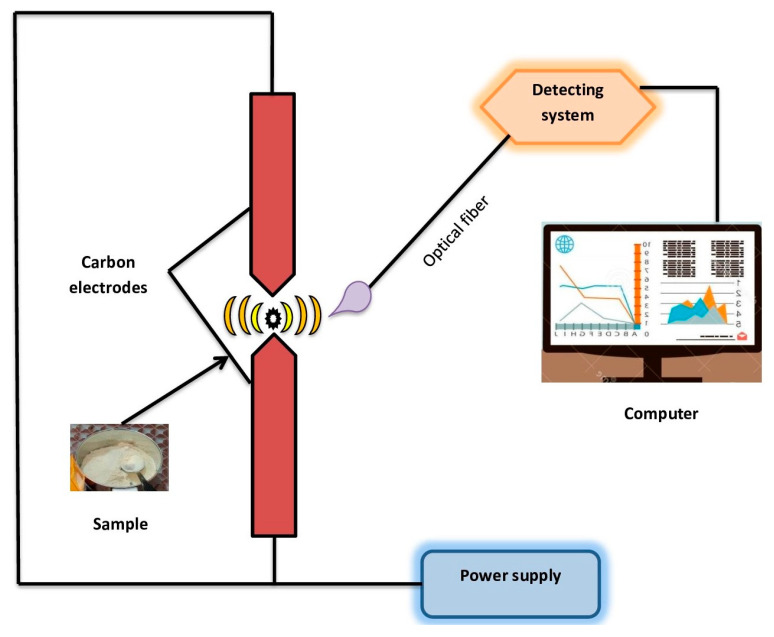
Schematic arrangement of direct current arc optical emission spectroscopy.

**Figure 2 metabolites-12-00821-f002:**
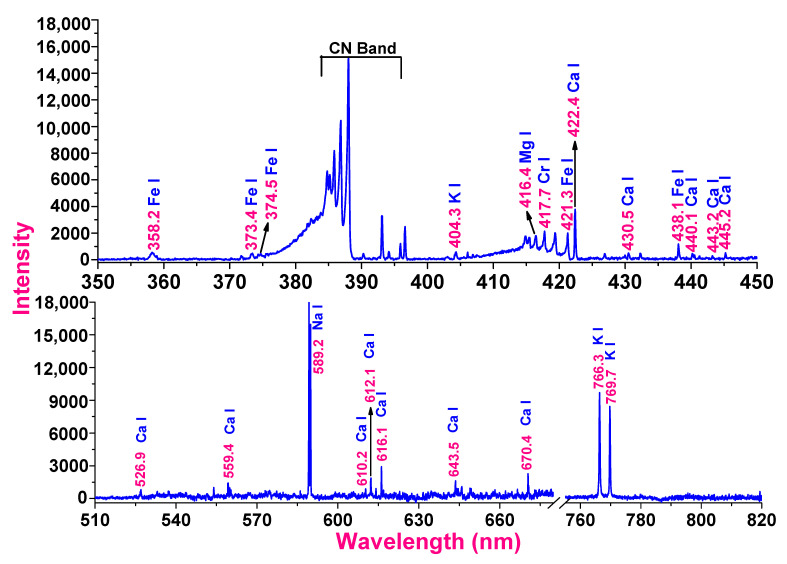
Recorded emission spectrum of the mushroom excited by direct current arc in the spectral region of 350–900 nm.

**Figure 3 metabolites-12-00821-f003:**
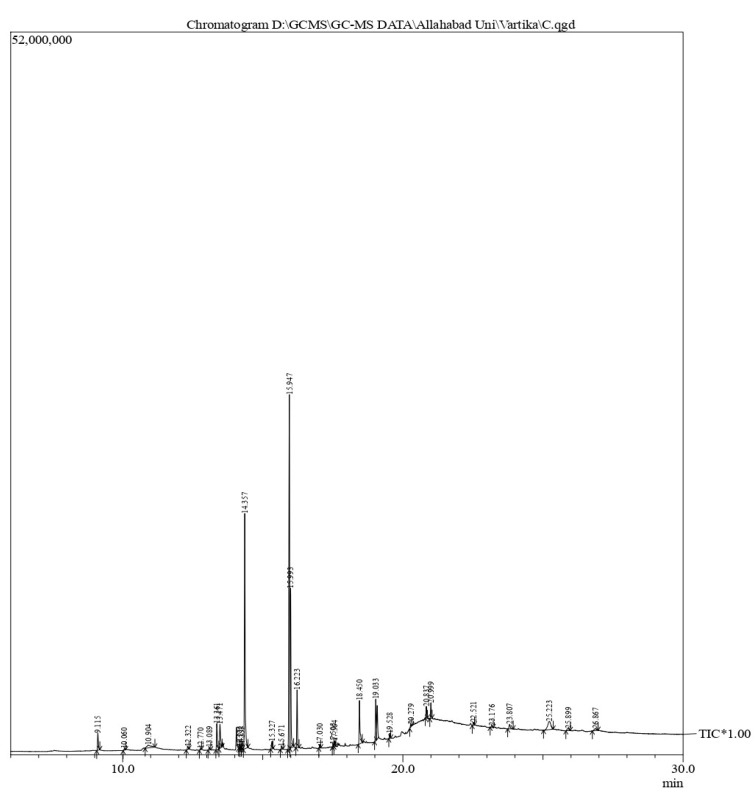
GC-MS chromatogram of the phytoconstituents present in the ethanol extract of *Pleurotus ostreatus* (EEPO).

**Figure 4 metabolites-12-00821-f004:**
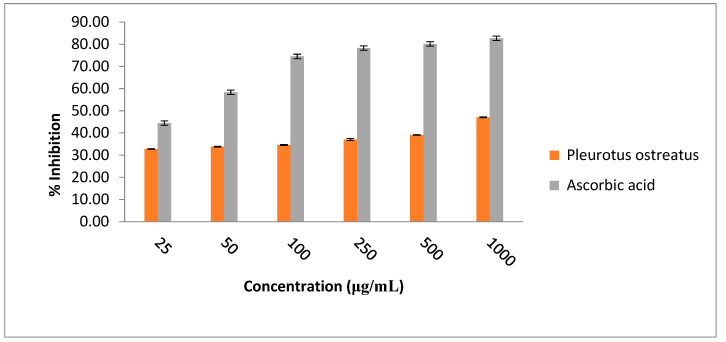
Concentration-dependent DPPH scavenging activities of ethanolic extract of *Pleurotus ostreatus*. Ascorbic acid used as reference.

**Figure 5 metabolites-12-00821-f005:**
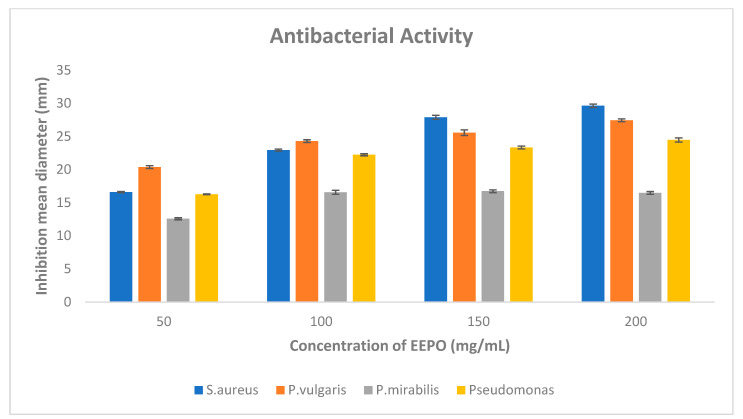
The agar well diffusion method was employed to determine the antibacterial activity of EEPO of various concentrations (50, 100, 150 and 200 mg/mL), and the averages of zones of inhibition in millimeters (mm) against Gram-positive and Gram-negative bacteria are graphically shown. Values are represented as mean ± SD.

**Figure 6 metabolites-12-00821-f006:**
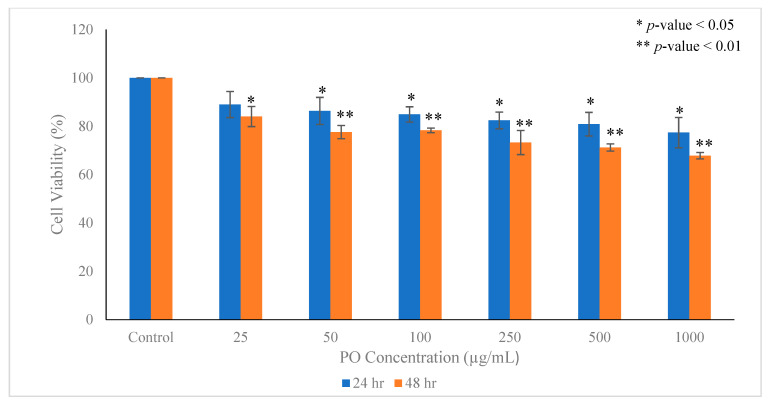
Statistical analysis of cellular viability using PrestoBlue and SRB assays after treating the cells with various concentration of *Pleurotus ostreatus* extract ranging from 25 to 1000 µg/mL at different time points (24 and 48 h). The results were analyzed using two-way ANOVA; data are expressed as mean ± standard deviation (SD), *n* = 3. The significance is represented as * *p* < 0.05, ** *p* < 0.01.

**Figure 7 metabolites-12-00821-f007:**
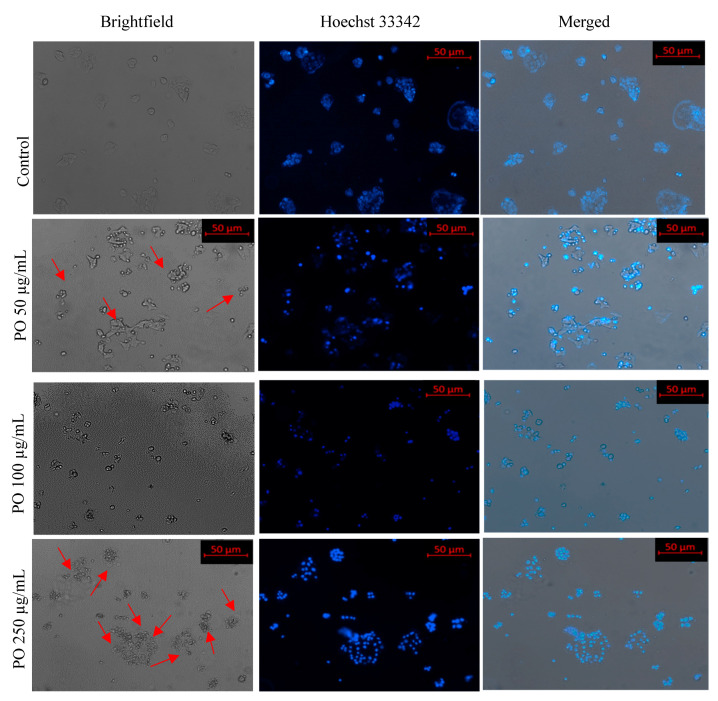
Nuclear morphology of MCF-7 cells after DAPI staining. The fluorescence microscopy images (at 10×) of cells treated with different concentrations of EEPOs show condensed nuclei, high fluorescence and cell shrinkage as compared to the negative control, which does not exhibit such nuclear changes. Red arrowhead: Cell apoptosis in treatment group.

**Figure 8 metabolites-12-00821-f008:**
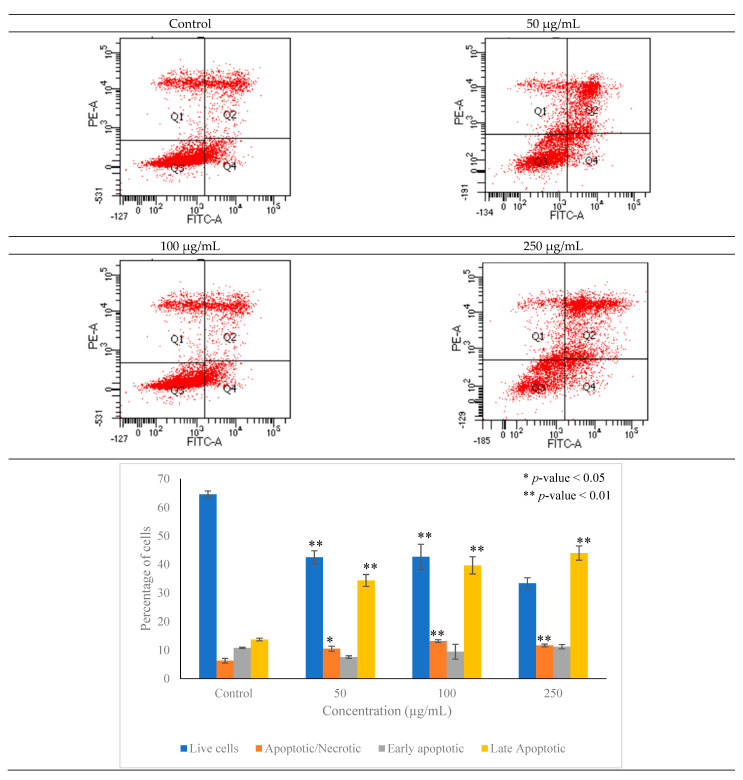
Cellular apoptosis evaluated using Annexin V/PI double staining by flow cytometry. The apoptosis rate of cells was significantly increased in cells treated with EEPO as compared to control cells. Apoptotic cell percentage was quantified after treating the cells with various concentrations of EEPO ranging from 50 to 250 µg/mL. The results were analyzed using two-way ANOVA; data are expressed as mean ± standard deviation (SD), *n* = 3. The significance is represented as * *p* < 0.05, ** *p* < 0.01.

**Table 1 metabolites-12-00821-t001:** Twenty-Five Available 3D Structures of Compounds from PubChem Database and 3D Structures of Selected Target Proteins from Protein Data Bank (PDB).

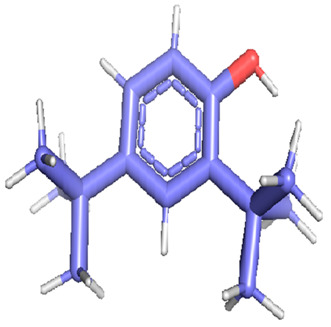	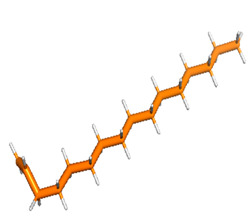
Phenol, 2,4-bis (1,1-dimethylethyl)-	1-Hexadecene
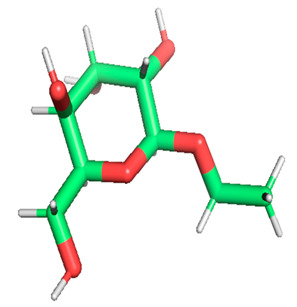	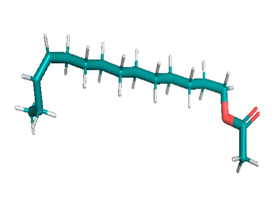
Ethyl. alpha. -d-glucopyranoside	11-Tetradecen-1-ol, acetate, (Z)-
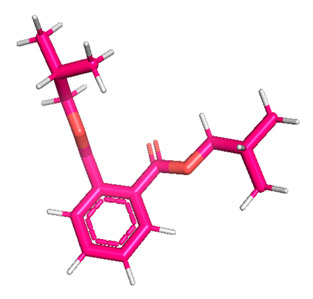	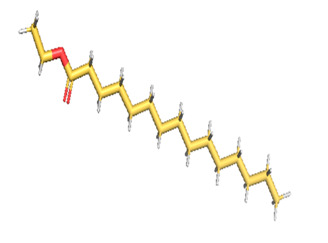
1,2-Benzenedicarboxylic acid, bis(2-methylpropyl) ester	Pentadecanoic acid, ethyl ester
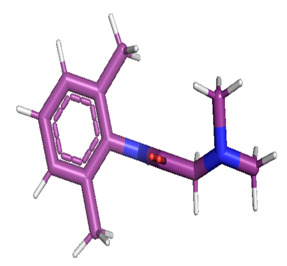	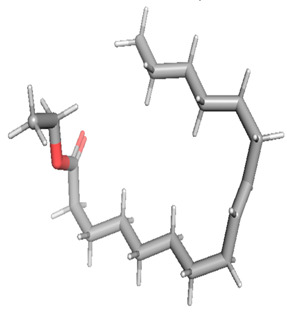
2-(Diethylamino)-N-(2,6-dimethylphenyl)ACE	Ethyl 9-hexadecenoate
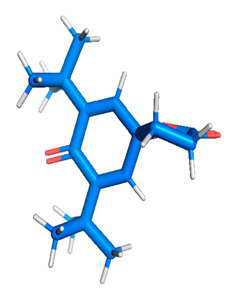	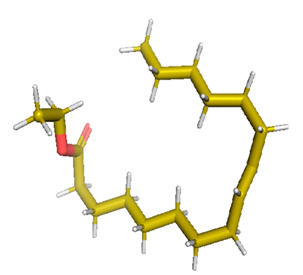
7,9-Di-tert-butyl-1-oxaspiro (4,5) deca-6,9-diene-2,8-dione	Ethyl 9-hexadecanoate
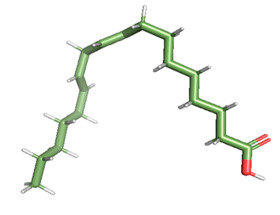	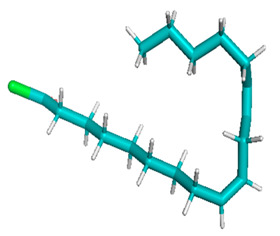
Hexadecanoic acid, ethyl ester	9,12-Octadecadienoic acid (Z,Z)-, methyl ester
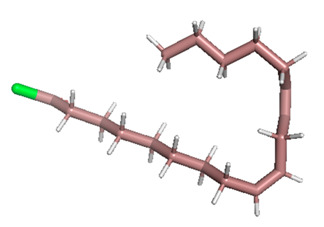	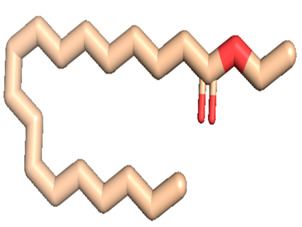
9,12-Octadecadienoyl chloride, (Z, Z)-	Linoleic acid ethyl ester
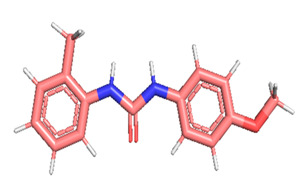	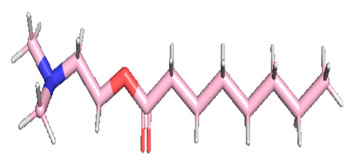
Octadecanoic acid, ethyl ester	Octanoic acid, 2-dimethylaminoethyl ester
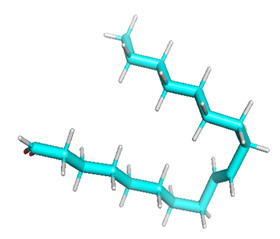	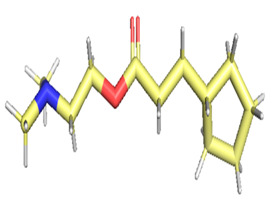
9-Octadecenal, (Z)-	3-Cyclopentylpropionic acid, 2-dimethylaminoethyl ester
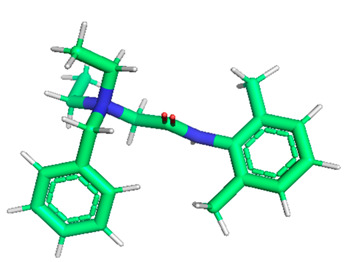	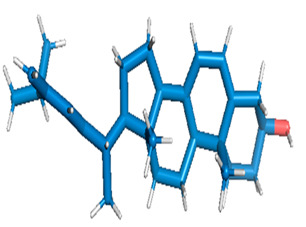
Benzyl diethyl-(2,6-xylylcarbamoylmethyl)-ammonium ben	Ergosta-5,7,9(11),22-tetraen-3-ol, (3. beta.,22E)-
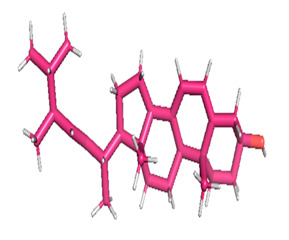	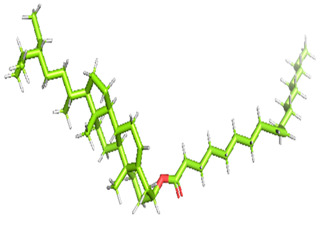
Stigmasta-5,22-dien-3-ol, acetate, (3. beta.,22Z)-	Stigmast-5-en-3-ol, oleat
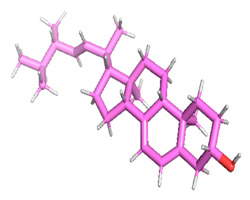	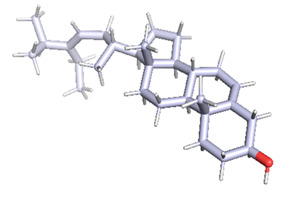
Ergosta-5,7,22-trien-3-ol, (3. beta.,22E)-	Stigmasta-5,23-dien-3-ol, (3. beta.)-
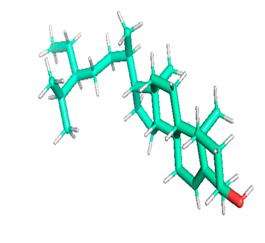	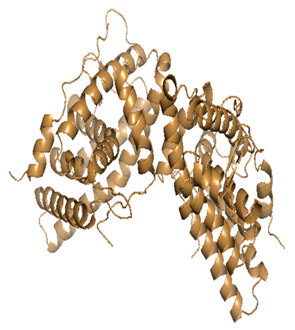
Gamma—Sitosterol	Progesterone, 1A28
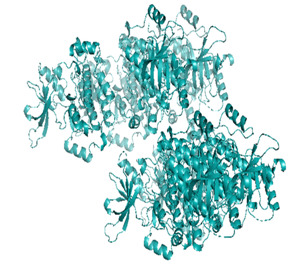	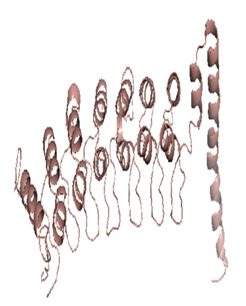
EGFR, 5GTY	NF-kB2, 4OT9

**Table 2 metabolites-12-00821-t002:** Gas chromatography–mass spectrometry (GC-MS) data of the bioactive compounds identified in this study. These bioactive compounds were identified from the ethanolic extract of *Pleurotus ostreatus* (EEPO).

Peak	R.Time	Area	Area%	Name	Ligand Name for Molecular Docking	Biological Activity	Ref.
1	9.115	2,518,815	1.66	Phenol,2,4-Bis(1,1-Dimethylethyl)	Ligand 1	Anticancer, antimicrobial and antioxidant	[11,12,13]
2	10.060	238,480	0.16	1-Hexadecene	Ligand 2	-	
3	10.904	2,087,528	1.38	Ethyl. alpha. -d-glucopyranoside	Ligand 3	Antioxidant and anticancer	[14]
4	12.322	391,627	0.26	9-Octadecenoic acid(Z)-, ethyl ester	Ligand 4	Anticancer	[15]
5	12.770	136,985	0.09	11-Tetradecen-1-ol, acetate, (Z)	Ligand 5	Used in cosmetics	[16]
6	13.089	451,454	0.30	1,2-Benzenedicarboxylicacid, bis(2-methylpropyl) ester	Ligand 6	Anticancer, apoptotic, antimicrobial and antifouling activity	[17,18]
7	13.361	3,516,405	2.32	Pentadecanoic acid, ethyl ester	Ligand 7	Antibacterial and antifungal	[19]
8	13.471	4,288,819	2.83	2-(Diethylamino)-N-(2,6-Dimethylphenyl) Ace	Ligand 8	-	
9	14.153	104,349	0.07	Ethyl9-hexadecenoate	Ligand 9	Antioxidant and anti-androgenic	[20]
10	14.211	170,369	0.11	7,9-Di-tert-butyl-1-oxaspiro (4,5) deca-6,9-diene-2,8-dione	Ligand 10	Antineoplastic, antimicrobial and antiviral	[21,22]
11	14.251	79,393	0.05	Ethyl9-hexadecanoate	Ligand 11	-	
12	14.357	32,499,008	21.42	Hexadecanoic acid, ethyl ester	Ligand 12	Anti-inflammatory, antioxidant, hypocholesterolemic, nematicide, pesticide, lubricant and anti-androgenic	[23,24]
13	15.327	1,174,559	0.77	9,12-Octadecadienoicacid (Z, Z)-, methyl ester	Ligand 13	Antibacterial, antifungal, antioxidant and anticancer	[18,20,23]
14	15.671	312,974	0.21	9,12-Octadecadienoylchloride, (Z, Z)	Ligand 14	Antisecretory, contraceptive, antitubercular and anti-spermatogenic	[25]
15	15.947	49,748,008	32.78	Linoleic acid ethyl ester	Ligand 15	Hypocholesterolemic, nematicide, antiarthritic, hepatoprotective, antiandrogenic, hypocholesterolemic, 5-alpha reductase inhibitor, antihistaminic, anti-coronary, insectifuge, anti-eczemic and anti-acne	[26]
16	15.993	20,864,605	13.75	(E)-9-Octadecenoicacidethylester	Ligand 16	Antimicrobial	[26]
17	16.223	7,107,650	4.68	Octadecanoic acid, ethyl ester	Ligand 17	5-alpha reductase inhibitor, hypocholesterolemic and lubricant	[26]
18	17.030	415,993	0.27	Octanoicacid,2-dimethylaminoethylester	Ligand 18	-	
19	17.503	309,994	0.20	Urea, N-[4-(diethylamino)-2-methoxyphenyl]-N’-(1,1-dimet	Ligand 19	-	
20	17.564	629,878	0.42	9-Octadecenal, (Z)	Ligand 20	Antimicrobial	[27]
21	18.450	6,818,740	4.49	3-Cyclopentylpropionicacid,2-dimethylaminoethylester	Ligand 21	Antimicrobial and antifouling	[16]
22	19.033	5,791,604	3.82	Benzyl diethyl-(2,6-xylylcarbamoylmethyl)-ammonium	Ligand 22	-	
23	19.528	600,854	0.40	Docosanoic acid, ethyl ester	Ligand 23	Cosmetics	[20]
24	20.279	328,001	0.22	Octadecanoic acid, ethyl ester	Ligand 24	5-alpha reductase inhibitor, hypocholesterolemic, lubricant and flavor	[26]
25	20.837	1,373,935	0.91	cis-15-Tetracosenoicacid, propyl ester	Ligand 25	Antimalarial activity	[28]
26	20.999	1,833,453	1.21	Ethyl tetracosanoate	Ligand 26	-	
27	22.521	420,128	0.28	Ergosta-5,7,9(11),22-tetraen-3-ol, (3. beta.,22E)	Ligand 27	-	
28	23.176	354,005	0.23	Stigmasta-5,22-dien-3-ol, acetate, (3. beta.,22Z)	Ligand 28	Antibacterial and antifungal	[29]
29	23.807	1,000,313	0.66	Stigmast-5-en-3-ol, Oleat	Ligand 29	Antihyperlipidemic and anti-tumor	[30]
30	25.223	4,663,969	3.07	Ergosta-5,7,22-Trien-3-ol, (3. Beta.,22E)	Ligand 30	Not reported	
31	25.899	724,079	0.48	Stigmasta-5,23-Dien-3-ol, (3. Beta.)	Ligand 31	Not reported	
32	26.867	785,623	0.52	Gamma. Sitosterol	Ligand 32	Anti-hypercholesterolemic, antiviral (influenza), anti-inflammatory, anti-acne, antiprotozoal (leishmania), antibacterial and antioxidant	[31]
		151,741,597	100.00				

**Table 3 metabolites-12-00821-t003:** Antibacterial activity of EEPO. Results are recorded as mean ± standard deviation.

mg/mL	*Staphylococcus aureus*	*Proteus vulgaris*	*Pseudomonas aeruginosa*	*Proteus mirabilis*
50	16.6 ± 0.1	20.3 ± 0.2	16.2 ± 0.05	12.5 ± 0.1
100	22.9 ± 0.1	24.3 ± 0.2	22.2 ± 0.1	16.5 ± 0.3
150	27.9 ± 0.3	25.5 ± 0.4	23.3 ± 0.2	16.7 ± 0.2
200	29.6 ± 0.2	27.4 ± 0.2	24.4 ± 0.3	16.4 ± 0.2

**Table 4 metabolites-12-00821-t004:** Binding energies (kcal/mol) of GC-MS compounds docked with target macromolecules.

Target Macromolecule and Binding Energy (kcal/mol)			
Compounds	EGFR 5GTY	NF-kB 4OT9	PROGESTERONE 1A28
lig_1	−6.8	−4.8	−6.7
lig_2	−5.6	−3.8	−4.3
lig_3	−5.6	−4.9	−5.9
lig_5	−5.4	−3.9	−3.3
lig_6	−6.8	−4.9	−6.6
lig_7	−4.4	−3.1	−3.4
lig_8	−6	−4.5	−5.8
lig_9	−5.1	−3.1	−3.3
lig_10	−7.6	−5.9	−6.5
lig_11	−5.1	−3.3	−3.4
lig_12	−5.5	−3.5	−4.1
lig_13	−4.9	−3.1	−3
lig_14	−5.4	−3.4	−4.4
lig_15	−13.2	−10.4	−11
lig_17	−6.9	−5.2	−6.1
lig_18	−5.5	−3.8	−4.1
lig_20	−4.9	−2.2	−3.4
lig_21	−6.2	−4.6	−2.9
lig_22	−7.4	−6.3	−5.4
lig_27	−9.7	−6.1	−6.7
lig_28	−9.4	−5.8	−8.7
lig_29	−7.9	−6.6	−6.3
lig_30	−8.6	−6.4	−6.3
lig_31	−8.2	−5.2	−6.2
lig_32	−8.5	−4.9	−5.1

**Table 5 metabolites-12-00821-t005:** Residues involved in the interaction of three best fit ligands with target proteins.

S.No.	Protein	Top Three Compounds with the Lowest Binding Energy and Names of Residues of Proteins that Showed Close Contact with Ligands		
1	5GTY	Lig15 (THR766, LEU764, LYS721, MET742, ASP831)	Lig27 (LEU694, VAL702, PHE699)	Lig28 (VAL702, MET769, LEU768, ALA719, LEU820, PHE699)
2	4OT9	Lig15 (ARG471, VAL510, TYR509, HIS513)	Lig29 (LEU734, THR732, HIS695, PRO756, GLU755, ALA750, ALA749, LEU688, LYS689, THR753, MET754, ALA692, ILE694)	Lig30 (GLU699, ALA667, HIS731, ASN698, ASN669, HIS673, LEU736, LEU674, GLY677)
3	1A28	Lig15 (GLU807, LEU883, LEU880, LEU876, ASN879)	Lig27 LYS734, VAL730, ARG740, GLN747, ILE748, ILE751, ILE744, GLN752, VAL912, GLU911	Lig28 VAL698, VAL729, SER728, GLU695 MET692, HIS770, MET759, GLN725, TRP765, GLY762, ARG766, PRO696

## Data Availability

The data presented in this study are available in article.

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
