# Peer review of "Elemental Analysis, Phytochemical Screening and Evaluation of Antioxidant, Antibacterial and Anticancer Activity of Pleurotus ostreatus through In Vitro and In Silico Approaches"

_metabolites, 2022, doi:10.3390/metabo12090821_

Round 1

Reviewer 1 Report

The paper entitled “Elemental analysis, phytochemical Screening and evaluation of antioxidant, antibacterial and anticancer activity of Pleurotus ostreatus through in-vitro and in-silico approaches” is important and relevant. The language of manuscript is lucid, terse, scientific and very well focused. However, the following minor changes can improve the quality of the manuscript:

 1)    The authors should incorporate appropriate references for biological activities mentioned in Table 2.

2) There is variation in the concentrations for antioxidant activity in materials and methods and figure 4. This needs to be rectified.

3) Include references for Presto blue assay and antibacterial activity protocol (refer to materials and methods).

4) Concentrations of extract in Cell apoptosis and nuclear morphology assays should be mentioned in the experimental section.

5)Specify the magnification of microscopy images.

Reviewer 2 Report

The authors didn’t define ligands used in molecular docking anywhere in the manuscript. The description of ligands should be added.

The authors should also incorporate the residues involved in the interaction of the top 3 ligands with receptors.

Scale bars are missing in images. It should be corrected.

Please provide an elaborate diagrammatic representation of the work in graphical abstract.

p-values should be mentioned in the graphs.

The conclusion should be more precise and discuss future prospects if any.

Reviewer 3 Report

There are some comments for the authors:

1. The flow of the manuscript is not so easy to follow. The authors should revise the text, so easy to understand what is the messages they want to convey. 

2. English: Please revise the manuscript from a native English speaker/professional editing services.

3. Introduction: There are several studies reported earlier regarding the antibacterial, antiviral, antidiabetic, anticancer, antitumor and antioxidant potential of Pleurotus ostreatus. The authors must shed light on these studies and justify the novelty of their investigation, and what novel findings have been derived from this study. And what gaps the authors are trying to fill in?

4. The significance of the central claims must be clarified in the context of the existing literature, what the present study adds to what was already done. And what gaps are the authors trying to fill-in?

5. Conclusion: The conclusion in the abstract is too general, please make more concise and related to the data obtained in the results.
